# An Umbrella Review of the Evidence of Sex Determination Procedures in Forensic Dentistry

**DOI:** 10.3390/jpm12050787

**Published:** 2022-05-13

**Authors:** João Albernaz Neves, Nathalie Antunes-Ferreira, Vanessa Machado, João Botelho, Luís Proença, Alexandre Quintas, Ana Sintra Delgado, José João Mendes

**Affiliations:** 1Clinical Research Unit (CRU), Centro de Investigação Interdisciplinar Egas Moniz (CiiEM), Egas Moniz, CRL, 2829-511 Monte de Caparica, Portugal; vmachado@egasmoniz.edu.pt (V.M.); jbotelho@egasmoniz.edu.pt (J.B.); anasintradelgado@gmail.com (A.S.D.); jmendes@egasmoniz.edu.pt (J.J.M.); 2Laboratório de Ciências Forenses e Psicológicas Egas Moniz (LCFPEM), Centro de Investigação Interdisciplinar Egas Moniz (CiiEM), Egas Moniz, CRL, 2829-511 Monte de Caparica, Portugal; natantfer@gmail.com (N.A.-F.); alexandre.quintas@gmail.com (A.Q.); 3Orthodontics Department, Egas Moniz Dental Clinic (EMDC), Egas Moniz, CRL, 2829-511 Monte de Caparica, Portugal; 4Quantitative Methods for Health Research (MQIS), CiiEM, Egas Moniz, CRL, 2829-511 Monte de Caparica, Portugal; luisfapro@gmail.com

**Keywords:** forensic dentistry, sex determination, sexual dimorphism, dental measurements, predictive models

## Abstract

Sex determination in forensic dentistry is a major step towards postmortem profiling. The most widely recognized method is DNA, yet its application in the dental field of forensic sciences is still impractical. Depending on the conditions of the remains, teeth are often the only surviving organ. Some systematic reviews (SRs) have been recently produced; hence this umbrella review critically assesses their level of evidence and provides an overall comprehensive view. An electronic database search was conducted in four databases (PubMed, Cochrane, Web of Science, and LILACS) and three grey search engines up to December 2021, focusing on SRs of sex determination through forensic dentistry procedures. The methodological quality of the SRs was analyzed using the measurement tool to assess SRs criteria (AMSTAR2). Five SRs were included, two of critically low quality and three of low quality. The SRs posited that canines are the most dimorphic teeth; oral tissue remnants are a rich source for sex determination by DNA tracing; and artificial intelligence tools demonstrate high potential in forensic dentistry. The quality of evidence on sex determination using dental approaches was rated as low. Well-designed clinical trials and high standard systematic reviews are essential to corroborate the accuracy of the different procedures of sex determination in forensic dentistry.

## 1. Introduction

Teeth are highly resistant to aggressive agents, each tooth piece morphology is peculiar, radiographic recording is routinely conducted in standard dental treatment, and the methodological analysis is technically easy and affordable [1,2,3].

Precise sex prediction is a key step for a postmortem forensic profile [4]. In standard dental measures, teeth have a high degree of sexual dimorphism [5,6,7,8]. Usually, male teeth are bigger than female teeth; nevertheless, data is not consensual and reverse dimorphism also takes place [6,9]. Sexual dimorphism may alter between different populations, due to the environment, available food resources, or genetic pool [9,10].

Forensic identification of an individual’s remains can be executed using dental methods, such as the palatal rugae [11], the development stage of third molars [12], odontological measures [2,4,6], or alternative sources of DNA [13]. Given the applicability of these strategies, dental-based forensic methods have been gaining some importance in forensic identification.

In comparison with other nonmetric and metric measurements, DNA analysis has greater accuracy in sex determination and was consequently the standard method [14]. Therefore, odontometric measures are only truly valid and useful when sex organs or other sexual characteristics are not present [5].

As a result, this umbrella review intended to evaluate the existing evidence on sex determination procedures in forensic dentistry. Our focus was two-fold: to ascertain its quality of evidence and the overall clinical accuracy of each procedure.

## 2. Materials and Methods

Preferred Reporting Items for Systematic Reviews and Meta-Analyses (PRISMA) guidelines were followed [15] (Appendix A) as well as the guide for systematic reviews of systematic review [16]. The research question was defined: “What is the current evidence on sex determination approaches in forensic dentistry?”.

### 2.1. Eligibility Criteria

To answer the proposed research question, the inclusion criteria were: (1) systematic review (with or without meta-analysis); (2) addressing sex determination using a forensic dentistry approach. No restrictions on language or publication year were applied.

### 2.2. Information Sources Search

Electronic data search was performed in four electronic databases: PubMed, Cochrane Database of Systematic Reviews, LILACS (Latin-American scientific literature in health sciences), and Web of Science up to December 2021. Keywords and subject headings were merged with the following syntax “((sex determination [MeSH terms]) OR (sex determination) OR (sex estimation) OR (gender estimation) OR (sex prediction) OR (canine index) OR (sexual dimorphism) OR (sex determination) OR (dental dimorphism) OR (gender determination) OR (sex dimorphism [MeSH terms]) OR (sex dimorphism)) AND (tooth OR teeth OR canine OR premolar OR molar) AND ((Systematic Review) OR (Meta-analysis))”. Grey literature was also scanned.

### 2.3. Study Selection

Two researchers (J.A.N. and J.B.) separately screened titles and abstracts. The agreement between the reviewers was assessed by Kappa statistics. Any paper classified as potentially eligible by either reviewer was ordered as a full-text and independently screened. All disagreements were solved through discussion with a third reviewer (V.M.).

### 2.4. Data Extraction Process and Data Items

Two researchers (J.A.N. and J.B.) separately extracted: authors and year of publication, objective/focused question, databases searched, number of studies included, type of studies included, main results, and main conclusions. All disagreements were solved through discussion with a third reviewer (V.M.). 

### 2.5. Risk of Bias and Methodological Quality Assessment

Two researchers (J.A.N. and J.B.) applied the measurement tool to assess systematic reviews (AMSTAR 2) to establish the methodological quality of the included reviews [16]. AMSTAR 2 is a comprehensive 16-item tool that rates the overall confidence of the results of the review. As stated in the AMSTAR guidelines, the quality of the systematic reviews was classified as follows: high means ‘zero or one non-critical weakness’; moderate means ‘more than one non-critical weakness’; low means ‘one critical flaw with or without non-critical weaknesses’; and critically low means ‘more than one critical flaw with or without non-critical weaknesses’. The estimation of the AMSTAR quality rate for each study was calculated through the AMSTAR 2 online tool (https://amstar.ca/Amstar_Checklist.php, accessed on 1 December 2021). 

### 2.6. Synthesis of Results

We anticipated a heterogeneous variety of systematic reviews, and for this reason meta-analysis was deemed not possible to carry out. Yet, we described the results of each systematic review according to the type of procedure: dental, molecular, and artificial intelligence methods, following the SWiM guideline [17].

## 3. Results

### 3.1. Study Selection 

Electronic searches redeemed a total of 370 titles through the database searching. Following manual assessment of title/abstract and removal of duplicates, 12 potentially eligible full texts were screened. Full-text screening excluded 7 studies with reasons, resulting in 5 systematic reviews that fulfilled the inclusion criteria (Figure 1). Inter-examiner reliability at the full-text screening was recorded as high (kappa score = 1.00).

### 3.2. Study Characteristics

In total, five systematic reviews were included in the present umbrella review (Table 1). All SRs covered a defined timeframe [14,18,19,20,21]. Three systematics reviews restricted their search to studies in English [18,20,21] while the remaining had no language restrictions [14,19].

#### Methodological Quality

Two studies were rated as of critically low quality [20,21] and three as of low quality [14,18,19] (detailed in Table 2). None of the included SR completely fulfilled the AMSTAR2 checklist. Overall, SRs mostly failed on: describing their selection of the study designs for inclusion (100%, *n* = 5); providing a list of excluded studies with the respective reasons for exclusion (100%, *n* = 5); accounting for RoB in individual studies when interpreting/discussing the results of the review (80%, *n* = 4); performing data extraction in duplicate (80%, *n* = 4).

### 3.3. Synthesis of Results

#### 3.3.1. Dental Methods

Overall, the level of evidence of the SRs regarding forensic tools based on dental methods was of low quality.

All human teeth exhibit a small degree of sexual dimorphism, with the canines and the second molars being the most dimorphic teeth [18,19,20]. Regarding canines mesiodistal measures alone, maxillary left canines were reported with the lowest sexual dimorphism. On the other hand, both right and left mandibular canines presented the higher dimorphic degree [19,20]. When considering all teeth, premolars showed the smallest dimorphic level [18,19,20].

Although dental dimorphism occurs among different racial groups, the difference between measures is very substantial, which seems to have geographically relevance due to environmental and genetic factors [19]. Nevertheless, it seems impossible to identify a universal predictor value across all populations [19,20].

The accuracy of methods based on measurements on casts was estimated to be between 34.5% and 90% (one study reported 100% accuracy) [18,19,20].

Despite biochemical analysis being presented as the most precise method, it is not appropriate in the forensic practice as an odontological sex estimation method [14].

#### 3.3.2. Molecular Analysis

The level of evidence on molecular analysis in forensic dentistry was of critically low quality.

Most samples considered were obtained from the dentin and the pulp because oral tissue remnants are a rich source for sex determination by DNA tracing, specially amelogenin, a protein of epithelial origin, and SRY or sex-determining region Y, a sex-typing based on the Y chromosome [21]. This type of sex determination could reach 100% of expected accuracy [21].

The available body and tissue remnants, as well as their level of degradation, drives the clinicians to make the appropriate methodological choice [21].

#### 3.3.3. Artificial Intelligence (AI) Technology in Forensic Dentistry

The level of evidence on AI-based methods for sex prediction using dental measures was of low quality.

In forensic dentistry, AI demonstrates accuracy and precision identical to qualified examiners, prevents possible human errors, and is non-invasive [18]. AI is a promising tool for mass disasters, especially in victim identification although there is an absence of real-life testing and validation [18].

## 4. Discussion

This umbrella review unequivocally summarizes the evidence sourced in sex determination in forensic dentistry. The methodological quality of the included SRs ranged from critically low to low quality, and consequently the present knowledge is supported by low-confidence evidence-based studies.

In forensic dentistry, sex determination is a key element during the process of human identification. It facilitates this process because the pool of missing people would be half, which allows the matching operation to be sped up [14]. Several studies were published regrading sex determination methods. Capitaneanu et al. [14] suggested that a reliable method should present an accuracy of over 80%, and even then, in a case of negative forensic match in the first search, it should be succeeded by a second search for antemortem data in both sexes [14]. Most studies tested Indian populations [14,19,20] but it is also stated that there are differences between populations, rendering the need for larger and geographically wider samples [19]. Many studies focus on a young adult population in order to ascertain the presence of healthy teeth, removing factors like tooth wear, cavities, and fractures from dental measurements [14,19]. Almost all studies agree with the election of the canine as the most dimorphic tooth, followed by the second molar [14,19,20]. On the other hand, premolar teeth showed the smallest dimorphism [14]. Few studies use skeletal remains, probably related to the difficulty of gathering a large sample of intact bodies with a full set of teeth [14].

As previously asserted, teeth are the hardest tissue in the human body, remaining intact during a considerable time and becoming a premium source for DNA collection, which is the most reliable source in sex determination. Samples were mostly collected from dental pulp, intraoral epithelial cells from swabs, toothbrushes, and saliva from dental prosthesis [21]. Age seems not to have an influence on sex determination via DNA analysis. DNA degradation took place in samples that underwent simulated environmental challenges, such as desiccation, incineration, submersion in saltwater, or burial. However, DNA can be successfully collected and processed for sex determination through DNA amplification techniques, such as polymerase chain reaction (PCR) [21]. The lack of standardization of the DNA sources and conditions prevents a correct interpretation of the results [21].

AI models mimic, most of all, the human brain, and its ability to create mathematical reasoning, problem-solving, and decision making. But to do so, these AI models must be taught and trained [18]. Concerning sex determination, AI was only applied to identify sexual dimorphism in canines using orthopantomography. The methods used were a multilayer perceptron and an artificial neural network, overperforming discriminant analysis and logistic regression, two standard sex determination methods [18]. AI will play a major role in forensic dentistry, not only in sex determination but also in several other fields, such as age determination, bitemarks, or lip prints [18].

With the purpose of improving the quality of research, the checklist from TRIPOD [22] should be followed. The research question should be clearer with special emphasis on selection of study designs and reasons for exclusion. It would also enrich future SRs if they took into account the RoB of individual studies as well as the number of authors that performed data extraction, to prevent bias.

### Strengths and Limitations

This umbrella review benefits from its comprehensive review of the available SRs using a clear methodology. However, one limitation does need to be accounted for when interpreting the results. In each SR, the individual studies included were not explored. Thus, the conclusions of this review are based on the interpretation of the authors.

## 5. Conclusions

The current knowledge is supported by low-confidence evidence-based studies. Although DNA is the best source for sex determination, forensic dental methods for sex estimation present some reliability. This umbrella review reports the most common mistakes performed in SRs and will pave the way for more robust evidence-based research in the future.

## Figures and Tables

**Figure 1 jpm-12-00787-f001:**
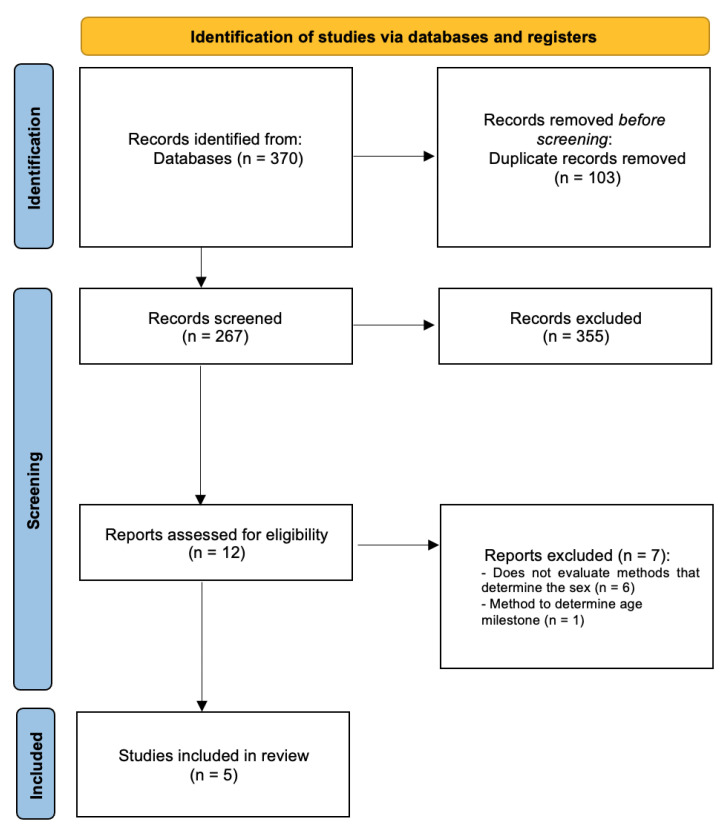
PRISMA flowchart of included studies.

**Table 1 jpm-12-00787-t001:** Characteristics of included SRs.

Authors (Year)	N	Search Period	Interventions	Quality Assessment Tool	Sample	Method of Analysis	Outcomes	AMSTAR2 Score *	Funding
Capitaneanu et al. (2017) [14]	103	Up to November 2016	Odontological sex estimation methods	QAS	NR	SR & MA	Accuracy	Low	NI
Khanagar et al. (2021) [18]	8	January 2000 to June 2020	AI-based models for sex determination, age estimation, and personal identification	QUADAS-2	NR	SR	Accuracy, sensitivity, specificity, ROC, AUC, ICC, PPV, NPV	Low	Research Grant
Maulani et al. (2020) [21]	10	2009 to 2019	DNA analysis methods	None	NR	SR	Accuracy	Critically Low	None
Pratapiene et al. (2016) [20]	11	January 2004 to April 2014	Canine mesiodistal measures	Cochrane Collaboration Tool	NR	SR & MA	Sexual dimorphism	Critically Low	NI
Silva et al. (2019) [19]	31	October 2015 to July 2016	Tooth crown mesiodistal measures	Criteria developed by the authors	NR	SR & MA	Sexual dimorphism	Low	NI

AI—artificial intelligence; DNA—deoxyribonucleic acid; MA—meta-analysis; N—number of included studies; NI—no information; NR—not reported; QAS—quality assessment tool. QUADAS—quality assessment and diagnostic accuracy tool; SR—systematic review. * Detailed information regarding the methodological quality assessment is present in Table 2.

**Table 2 jpm-12-00787-t002:** Methodological quality of the included SRs.

First Author	1	2	3	4	5	6	7	8	9	10	11	12	13	14	15	16	Review Quality
Capitaneau et al. (2017) [14]	Y	PY	N	PY	Y	N	N	Y	PY/PY	N	Y/Y	N	N	Y	N	Y	Low
Silva et al. (2019) [19]	Y	PY	N	PY	Y	N	N	Y	PY/PY	N	Y/Y	N	N	Y	N	Y	Low
Khanagar et al. (2020) [18]	Y	PY	N	PY	Y	Y	N	Y	PY/PY	N	0/0	0	N	N	0	Y	Low
Maulani et al. (2020) [21]	N	N	N	PY	N	N	N	PY	N/N	N	0/0	0	N	N	0	Y	Critically Low
Pratapiene et al. (2016) [20]	Y	PY	N	N	N	N	N	Y	PY/PY	N	Y/Y	Y	Y	N	N	Y	Critically Low

0—No meta-analysis conducted, N—No, Y—Yes, PY—Partial Yes. 1. Are research questions and inclusion criteria included? 2. Were review methods established a priori? 3. Is there an explanation of the review authors’ selection literature search strategy? 4. Did the review authors use a comprehensive literature search strategy? 5. Was study selection performed in duplicate? 6. Was data selection performed in duplicate? 7. Is the list of excluded studies and exclusions justified? 8. Is the description of the included studies in adequate detail? 9. Is there a satisfactory technique for assessing the risk of bias (RoB)? 10. Is there a report on the sources of funding for the studies included in the review? 11. If meta-analysis was performed, did the review authors use appropriate methods for statistical combination of results? 12. If meta-analysis was performed, did the review authors assess the potential impact of RoB? 13. Was RoB accounted for when interpreting/discussing the results of the review? 14. Did the review authors provide a satisfactory explanation for, and discussion of, any heterogeneity observed in the results of the review? 15. If they performed quantitative synthesis, was publication bias performed? 16. Did the review authors report any potential sources of conflict of interest, including funding sources?

## Data Availability

Not applicable.

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
