# Peer review of "An Umbrella Review of the Evidence of Sex Determination Procedures in Forensic Dentistry"

_jpm, 2022, doi:10.3390/jpm12050787_

Round 1

Reviewer 1 Report

The topic investigated is quite interesting for the forensic anthropology community. As stated by the Authors, different studies reported about sexual determination in human teeth and establishing specific reliability of different methods could help forensic analysis of human remains.

The strength of the manuscript could be that there are few publications that report the dental evaluation using methods to evaluate the quality of previous realized works. However, there are some inaccuracies which must be corrected before considering the publication (minor revision).

The Manuscript should also focus on anthropological sex estimation on other bones and not only comparing to DNA. Some deepening into AI technology in forensic dentistry could be interesting for the Reader in order to better understand what to expect for the future in this field.

English grammar should be checked.

Author Response

We would like to thank you for the feedback and the chance to improve our paper. Regarding sex estimation based on alternative bones, we would be keen to acknowledge such literature in the text however it goes beyond the scope of this review because the objective of our paper has focused Forensic Dentistry.
We have rewritten some segments on AI technology as suggested and now reads as follows:

AI will play a major role in Forensic Dentistry, not only in sex determination but also in several other fields, such as age determination, bitemarks or lip prints [18].

We also did a new review of the entire manuscript to check spelling and grammar.

Reviewer 2 Report

I think that this is a useful study because it gives an opportunity to see in one place the current situation with sex determination by use of dental forensic methods. The study is well conducted and that procedure is well explained. But I would suggest to the authors to rewrite some parts in abstract, introduction and discussion.

First, authors use term '' Forensic Dentistry'' as it is a special field in which sex determination can be used. But actually, FORENSIC DENTAL METHODS can often be used in FORENSIC practice. So think about changing this, so you can be more clear.

Similar , in the second sentence of abstract, authors have written that DNA methods are impractical in Dental fields. It is not clear what field of dentistry could need DNA methods.

Second,  in line 41 (introduction), authors said that development of third molar can be used for sex determination. It is true that time of apical closure of third molar differ in male and female but I think that this should be explained because it is not a big difference and like this it sounds like it is a good method (and it is not, is its just an additional remark form the authors).

Finally, I think authors should give some numbers of how accurate these dental methods for sex determination can be. This is important when some expert has to give a report for the court. So if he doesn't have any other method to determine the sex, he would choose a dental method, but he must also say how certain he is and give a reference. So you should provide this in your article, so we can use it sometimes.  

good luck!

Author Response

think that this is a useful study because it gives an opportunity to see in one place the current situation with sex determination by use of dental forensic methods. The study is well conducted and that procedure is well explained. But I would suggest to the authors to rewrite some parts in abstract, introduction and discussion.

Our Answer: We appreciate this overall commentary on our manuscript.

First, authors use term '' Forensic Dentistry'' as it is a special field in which sex determination can be used. But actually, FORENSIC DENTAL METHODS can often be used in FORENSIC practice. So think about changing this, so you can be more clear.

Our Answer: We thank you for this remark and the opportunity to improve our paper. The term “Forensic Dentistry” is one of the two terms mostly used nowadays of a specific field of Forensic Anthropology, being the other “Forensic Odontology”. We decided in previous papers to use “Forensic Dentistry” and it has never been a question raised before, since both terms are used in the systematic revisions included.

Similar , in the second sentence of abstract, authors have written that DNA methods are impractical in Dental fields. It is not clear what field of dentistry could need DNA methods.

Our Answer: The abstract has been altered as suggested to include “Dental field of Forensic Sciences…” for a better understanding.

Second,  in line 41 (introduction), authors said that development of third molar can be used for sex determination. It is true that time of apical closure of third molar differ in male and female but I think that this should be explained because it is not a big difference and like this it sounds like it is a good method (and it is not, is its just an additional remark form the authors).

Our Answer: Regarding line 41 (introduction), at any point we intend to state that third molar development can be used for sex determination. We state that forensic identification, which includes sex identication or age estimation, for instance, can be performed with dental methods, therefore justifying the importance of such umbrella review regarding solely sex determination.

Finally, I think authors should give some numbers of how accurate these dental methods for sex determination can be. This is important when some expert has to give a report for the court. So if he doesn't have any other method to determine the sex, he would choose a dental method, but he must also say how certain he is and give a reference. So you should provide this in your article, so we can use it sometimes. 

Our Answer: With respect to the accuracy of the different dental methods, it has been added to the Results section on each method, when referred by the studies included. Thank you for your valid recommendation.

Round 2

Reviewer 2 Report

Your work looks better now. Good luck!